# Synthesis, Structural Elucidation and Pharmacological Applications of Cu(II) Heteroleptic Carboxylates

**DOI:** 10.3390/ph16050693

**Published:** 2023-05-03

**Authors:** Shaker Ullah, Muhammad Sirajuddin, Zafran Ullah, Afifa Mushtaq, Saba Naz, Muhammad Zubair, Ali Haider, Saqib Ali, Maciej Kubicki, Tanveer A. Wani, Seema Zargar, Mehboob Ur Rehman

**Affiliations:** 1Department of Chemistry, Quaid-i-Azam University, Islamabad 45320, Pakistan; shakirwazir23@gmail.com (S.U.); afifa_chemist@hotmail.com (A.M.); sabanaz20014@gmail.com (S.N.); zubairmarwatqau@gmail.com (M.Z.); ahaider@qau.edu.pk (A.H.); 2Department of Chemistry, University of Science and Technology Bannu, Bannu 28100, Pakistan; zafranmwt35@gmail.com; 3Department of Chemistry, Adam Mickiewicz University in Poznan, 61-712 Poznan, Poland; maciej.kubicki@amu.edu.pl; 4Department of Pharmaceutical Chemistry, College of Pharmacy, King Saud University, P.O. Box 2457, Riyadh 11451, Saudi Arabia; twani@ksu.edu.sa; 5Department of Biochemistry, College of Science, King Saud University, P.O. Box 22452, Riyadh 11451, Saudi Arabia; szargar@ksu.edu.sa; 6PIMS Cardiac Center, Islamabad 44000, Pakistan; drmehboobfcps@yahoo.com

**Keywords:** substituted phenylacetic acid, Cu(II) carboxylates, DNA binding, enzymatic activity, antioxidant activity

## Abstract

Six heteroleptic Cu(II) carboxylates (**1**–**6**) were prepared by reacting 2-chlorophenyl acetic acid (**L^1^**), 3-chlorophenyl acetic acid (**L^2^**), and substituted pyridine (2-cyanopyridine and 2-chlorocyanopyridine). The solid-state behavior of the complexes was described via vibrational spectroscopy (FT-IR), which revealed that the carboxylate moieties adopted different coordination modes around the Cu(II) center. A paddlewheel dinuclear structure with distorted square pyramidal geometry was elucidated from the crystal data for complexes **2** and **5** with substituted pyridine moieties at the axial positions. The presence of irreversible metal-centered oxidation reduction peaks confirms the electroactive nature of the complexes. A relatively higher binding affinity was observed for the interaction of SS-DNA with complexes **2**–**6** compared to **L^1^** and **L^2^**. The findings of the DNA interaction study indicate an intercalative mode of interaction. The maximum inhibition against acetylcholinesterase enzyme was caused for complex **2** (IC_50_ = 2 µg/mL) compared to the standard drug Glutamine (IC_50_ = 2.10 µg/mL) while the maximum inhibition was found for butyrylcholinesterase enzyme by complex **4** (IC_50_ = 3 µg/mL) compared to the standard drug Glutamine (IC_50_ = 3.40 µg/mL). The findings of the enzymatic activity suggest that the under study compounds have potential for curing of Alzheimer’s disease. Similarly, complexes **2** and **4** possess the maximum inhibition as revealed from the free radical scavenging activity performed against DPPH and H_2_O_2_.

## 1. Introduction

The use of metal-based drugs as a therapeutic agent is evident from ancient times, uncovered some 5000 years ago, and they also form the basis of modern pharmacology [1]. The fortuitous discovery of cisplatin, its potency, and related side effects led to an increased research interest for the synthesis of new metal-based drugs [2]. Moreover, the failure of the already-in-use antibiotics in controlling diseases caused by microbes is considered to be the one of the most important issues by WHO, putting a great responsibility on the researchers in biological science for the discovery of potent and safe metallodrugs [3]. Mostly, the present emphasis is on the synthesis of drugs that target the DNA responsible for biochemical processes occurring in cells. Cisplatin, which exerts its effect by interacting with DNA, is the most inspiring example of metal-based drugs. However, its efficacy is severely affected by its toxic side effects [4]. Similarly, the synthesis of drugs capable of inhibiting enzymes that may help in terms of health and disease treatment has also been the focus of current research. The most promising enzymes, whose inhibition is considered helpful in the pathology of Alzheimer’s Disease (AD), may be acetylcholinesterase (AChE) and butyrylcholinesterase (BChE) [5]. AD is a major problem that is faced by developed countries with a high population of old age people [6]. Although there is no exact information as the possible causes of AD, an increase in the amount of acetylcholine as a result of inhibition of acetylcholinesterase is considered to be an effective strategy for the treatment of AD [7]. So, the drugs responsible for inhibition of these two enzymes are of growing interest; however, the already-in-use drugs are suffering from side effects and selective activities [7]. This puts a great demand on the researchers for the synthesis of effective, less toxic, and enzyme-targeting-drugs with a broad range of activities [6].

However, the synthesis of metal-based drugs with the required characteristics is not an easy task, as one has to be careful about the possible toxicity, the lethal effects of metal accumulation, unnecessary interaction with biomolecules and many more aspects [8]. Inspired from the natural biological macromolecules where a suitably organized complex architecture performs multitask functions, a huge amount of research has been focused on the synthesis of heteroleptic complexes, which offer great structural diversity [9]. The proper selection of metal and ligand is one of the most influential factors contributing towards the desired final geometry of complexes. The metal is considered to be the heart of coordination complexes, whereas the ligands exert their influence on physiochemical characteristics and applications as well [10].

Among the first row transition metals, Cu, which possesses biologically compatible chemistry, may be a good choice as metal center for the synthesis of complexes with a desired biological application. With the ability to adopt various easily accessible oxidation states, it is part of many enzymes involved in important biochemical processes in mammalian cells [11]. Moreover, being an essential trace metal, there is no fear of toxicity, and its concentration can be adjusted by the bio system. The use of copper for medicinal purposes is apparent from prehistoric times. It was used to sterilize wounds and water, treat chronic infections, kill fatal microbes and treat various diseases [12]. The complexation of copper with two different bioactive ligands can be enhanced further as of result of chelation and, hence, an increased lipophilic character [13,14].

The carboxylic acids are a good choice as a primary ligand and they can adopt a variety of interesting coordination modes, which assist complexes to adopt biologically suitable fascinating topologies. Besides this, other characteristic features such as acidity and the ability to develop electrostatic and hydrogen bonding allow them to interact with the target [15]. A number of carboxylic acids, especially the derivatives of aromatic carboxylic acids like phenyl acetic acid, already display their role as anti-inflammatory, antipyretic, and antitumor agents. The substituted phenylacetic acids are the natural ingredients of plant and fruits and are added in cosmetics and foods to induce flavors and fragrances. They also play important pharmacological roles, as they are used as precursors for the synthesis of clinically employed drugs, virostatic agents, pain-relieving agents, etc. [16]. However, there are limitations for the use of carboxylic acids as drugs due to lability, toxicity resulting from metabolism, and reduced bioavailability as a result of its restricted ability to cross the cell membrane [17]. The attached metal center and nitrogen donor heterocycling as an auxiliary ligand will not only help to overcome these limitations but also add to the coordination flexibility and structural diversity. These elements assist each other in order to achieve the desired qualities via extended π system, various supramolecular interactions and extended chelation [18].

Most of the commonly employed drugs such as Nonsteroidal anti-inflammatory agents (NSAIDs) are derivatives of carboxylic acids and, compared to free precursors, possess enhanced bioactivities on complexation with a suitable metal center and additional nitrogen donor co-ligand (such as pyridine and its derivatives [19,20,21]). With the introduction of auxochromes like –CN, -OH, and –NO_2_, the electron acceptance and fluorescence properties can be readily tuned. Cyanopyridine moiety, which is the most versatile organic intermediate, possesses an electron-withdrawing cyano group over an electron-accepting pyridine ring. The heteroleptic Cu(II) carboxylates with nitrogen donor heterocycles acting as auxiliary ligands have been characterized and found to show enhanced pharmacological potency [22,23].

Keeping in view the current demand as well as the relationship between structural diversity and biological significance, six new heteroleptic Cu(II) carboxylates were synthesized by using substituted phenylacetic acid as the primary ligand and substituted pyridine as the auxiliary ligand. They were characterized structurally and were evaluated for their DNA binding interaction through multi-spectroscopic techniques as well as for other pharmacological applications.

## 2. Results and Discussion

The heteroleptic Cu(II) carboxylates derived from the substituted chlorophenyl acetic acid were obtained in a good yield and pure form reflected from the values of their melting points. The heteroleptic carboxylates show solubility in the DMSO, ethanol, and methanol.

### 2.1. FT-IR Study

The broad bands appearing around 3400–3000 cm^−1^ originating from the –OH of carboxylate moiety of ligand acids **L^1^**–**L^3^** were absent in the spectra of synthesized Cu(II) carboxylates, thus indicating deprotonation of the acids (Table 1). The appearance of new peaks in the finger print region in the spectra around 500 cm^−1^ attributed to the Cu-O bond further support the deprotonation and complexation [24]. Bands of vibrational energy coming from the carboxylate moiety’s C-O bond vanish, whereas bands originating from the COO bond break into two bands, which are then classified as symmetrical around the COO and asymmetrical around the other atoms [24]. These two peaks are of great significance in the vibrational spectra of carboxylates and the difference in their values, i.e., Δν, is indicative of possible coordination modes adopted by the carboxylate moiety. It has been suggested for a number of similar reported dinuclear Cu(II) carboxylates with paddle wheel structures that a Δν falling in between 170–250 cm^−1^ indicates bridging bidentate coordination [24]. Here, too, for all six complexes, the values of Δν fall in the range of 185 cm^−1^ to 215 cm^−1^, suggesting the presence of a bridging bidentate coordination mode adopted by the carboxylate moiety. This suggestion is also supported from the single crystal XRD data of the complex **2** and **5**, where the existence of this kind of coordination results in a paddle wheel structure. The attachment of the substituted pyridine moieties in the complexes is evident from the appearance of vibrational bands responsible for Cu-N bond around 500–550 cm^−1^ [25]. Furthermore, the vibrational band around 2200 cm^−1^ attributed to the stretching of the cyano group (C≡N) of 3-CNPy and 4-CNPy also confirms the presence of these moieties [26]. The vibrational bands arising from the stretching of the Ar-Cl group appear in their respective region around 720–760 cm^−1^ in the spectra of all complexes without any shift, indicating the presence of ligand moieties and the non-involvement of the substituted Cl group in bonding [27].

### 2.2. Single Crystal XRD Analysis

The essential experimental details for the crystals of complexes **2** and **5** are given in Table 2, whereas the data for selected bond lengths and bond angles are given in Table 3. Figure 1 and Figure 2 show the solid-state structures of both complexes. Both **2** and **5** are *C_i_*-symmetrical (occupying inversion centers in their space groups) neutral di-nuclear complexes, in which two Cu(II) ions are bridged by four *syn*, *syn*-*η*^1^:*η*^1^:*μ* carboxylates, showing a paddle-wheel cage type structure. The coordination environment around each copper is a (CuNO_4_) distorted square pyramid as suggested for previously reported similar complexes [28,29]. The values of Cu-O_eq_ distance (Table 3), ranging from 1.962(1) to 1.975(1) Å in **2** and from 1.962(1) to 1.976(1) Å in **2**, are in close agreement as proposed for similar reported dinuclear Cu(II) carboxylates. The value of Cu…Cu bond distances 2.6061(3) Å in **1** and 2.6236(4) Å in **2**, which is less than the sum of the van der Waal’s radii of 2.8 Å, are also in close agreement with previously reported Cu(II) carboxylates coordinating to apical ligands which have an N-donor atom [29,30,31,32]. The Cu–N axial bond lengths are 2.159 Å in **1** and 2.165(2) Å in **2** and are attributed to the elongation of apical Cu–ligand bond distance as a consequence of repulsion exerted by the doubly occupied dz^2^-orbital along this axis, i.e., the Jahn–Teller effect. The elongated Cu-N bonds as compared to the Cu-O are attributed to the bigger covalent radii of nitrogen leading to distortion [33]. The distortion in geometry is also evident from the values of cisoid and transoid angles which are in the range of 84.13(6)°–97.33(6)° and 169.28(6)°–171.22(5)° in **1** and 83.43(4)°–99.38(6)° and 168.93(6)°–174.85(5)° in **2**, respectively, deviating from the values of 90° and 180° as prescribed for the ideal square pyramidal geometry [29].

### 2.3. UV-Visible Absorption Spectroscopy

The UV–Visible spectroscopic study of the ligand **L^1^**, **L^2^** and complexes **1**–**6** was performed in absolute ethanol and the absorption spectra for representative **L^1^**, and its complexes are given in Figure 3 whereas those for **L^2^** and its complexes are given in Appendix A. The ligand acids **L^1^** and **L^2^** both show maximum absorption in the region 266–277 nm, owing to the intra-ligand π-π* transitions in the aromatic system. In complexes **1**–**6**, this ligand-based absorption maxima shows a shift in wavelength due to the increased conjugation resulting from the formation of new rings [34]. It is described in the single crystal XRD that the complexes possess a distorted octahedral geometry for which the highest energy orbital is *d_z_*^2^, so three different transitions, i.e., *d_z_*^2^ → *d_x_*^2^_-*y*_^2^, *d_xy_* → *d_z_*^2^, *d_xz_*, *d_yz_* → *d_xy_* are expected. However, the spectra of complexes **1**–**6** comprise only a single broad band. This is because all four *d* orbitals lie very close to each other and precise assignment of each d-d transition is a difficult task, as the order of d energy orbitals is controversial among several researchers. Here, too, the absorption peak only get broadened [35].

### 2.4. Electrochemical Study

A 3 mM solution of ligand acids **L^1^**, **L^2^** and their complexes **1**–**6** in DMSO were used to record their voltammograms at a scan rate of 100 mv/sec; the data are given in the Appendix A. The cyclic voltammograms for complex **2** and **5** are given in Figure 4, whereas the cyclic voltammograms for **L^1^**, **L^2^**, **1**, **3**, **4**, and **6** are given in Appendix A. The voltammograms of the ligand acids indicate that the ligands were electrochemically silent at different scan rates. Meanwhile, the voltammograms of the complexes **1**–**6** reveal the presence of the redox couple, independent oxidation, and reduction peaks attributed to the presence of electroactive moieties present in the structural motifs of the complexes **1**–**6**. Here, for the complexes **2** and **5** with 4-cyanopyridine moiety as auxiliary ligand, two redox couples (in addition to one independent oxidation and one reduction potential) were observed. Similar behavior is observed for the rest of the complexes except that the second redox couple is missing and is replaced with two broad reduction peaks. The literature review concerning the substituted pyridine reveals that, in the case of 4-cyanopyridine, the complex gives a voltammogram with two redox couples, one at the positive side and other at the more negative side. Similarly, for 3-cyano pyridine, the first redox couple in the positive direction is dominant, whereas the second one on the negative side is diminished and left with only a broad reduction peak [36]. Similarly, dimeric Cu(II) carboxylates with the structural motifs similar to those described here produce electrochemical response with two redox couples: one attributed to inter-conversion of C(II)/Cu(III) and the second one originates from the reduction of Cu(II) to Cu(I) and vice versa [37,38,39]. So, based on the above description, it may be concluded that, for complexes **1**–**6**, the first very broad redox couple has originated as a result of overlap between the oxidation and reduction potentials associated with the metal center auxiliary ligands, i.e., substituted pyridine. In the voltammograms of the complexes **2** and **5**, an additional redox couple is observed at the very negative potential, which may be attributed to the 4-cyanopyridine moiety as per cited literature [36]. The other independent oxidation and reduction peaks in the voltammograms of the complexes may be attributed to the transition in heterocyclic moieties and may prove useful in the biological study [40].

### 2.5. DNA Interaction

#### 2.5.1. Through UV-Visible Absorption Spectroscopy

In order to get an idea about the mode and extent of interaction of the compounds with SS-DNA, the absorption spectra were recorded during the incremental addition of aqueous solution of DNA to the constant concentration of ligand acids and complexes. The representative absorption spectra of complexes **2** and **6** are given in Figure 5, whereas those of the **L^1^**, **L^2^**, **1**, **3**, **4** and **5** are given in Appendix A. Mostly, the small molecules interact with DNA irreversibly through non-covalent interaction which may involve intercalation, electrostatic interaction, and surface binding through major or minor grooves. Modality of interaction was determined by monitoring absorption spectra for variations (hypo-, hyper-chromic effect, red, or blue shift) with the incremental addition of DNA. In the present study, for the ligand acid as well as for the synthesized complexes **1**–**6**, it was found that there was a significant hypochromic impact, along with a highly red shift of around 3 nm to 5 nm. This shift in absorption maxima and absorption intensity is the result of change in electronic transitions which depend on the number, alignment, and distance between the compounds under study and the chromophore of DNA. It is expected that the intercalating moieties sitting (insertion) in between the adjacent DNA base pairs, π–π stacking between the base pairs, and aromatic ring system stabilized this insertion. In order to avoid any expected distortion, space is provided to the intercalating moiety by the separation of base pairs of DNA to some extent, i.e., the DNA unwind, leading to an increased distance between adjacent phosphatase and thereby an increase in the length of DNA duplex [41]. This alters the absorption spectra and results in a bathochromic effect (red shift). This is because of decrease in π-π* electronic transition energies due to the overlap between orbitals of intercalating moiety and base pairs of DNA. Moreover, as π* of the compound is partially filled with electrons from DNA base pair, there is a decrease in electronic transition, which results in a decrease in absorption, i.e., hypochromism. Based on these changes, the intercalative mode of interaction is expected for the ligand acids **L^1^**, **L^2^** and complexes **1**–**6**, which is also in accordance with the similar reported Cu (II) carboxylates. All of this is suggestive that the structural moieties of the compounds under study and DNA are in a direct bonding connection. They together with other structural changes are capable of altering the repairing process of DNA and hence prove to be effective against a disease [42].

A plot between A°/A-A° vs. 1/(DNA) of the Benesi–Hildebrand equation was used for the determination of binding constant. The intercept-to-slope ratio of these graphs was used to calculate binding constants K (M^−1^), which are given in Table 4. The binding constant values are high compared to the already reported complexes and may be attributed to the planar moiety and active substituents on the aromatic ring.

#### 2.5.2. Through Cyclic Voltammetry

To corroborate the findings from UV-visible spectroscopy on the contact behavior of ligand acids and complexes with SS-DNA, an electrochemical study was conducted. The cyclic voltammograms of complexes (ligand acids were found to be electrochemically innocent) were recorded during an incremental addition of DNA (0.49, 0.99, 1.48, 1.96 µM) to the constant concentration of complexes (3 mM) under the electrochemical setup earlier explained in Section 3.3 (3 mM). The voltammograms of complexes **1**, **2**, **5**, and **6** are given in Figure 6, whereas those of the complexes **1**, **3**, **4**, and **6** are presented in Appendix A. Different scan rates (50–500 V) were used to record the electrochemical response. Recurring changes in the peak potential and intensification of the current are hallmarks of irreversible electrochemical processes. Because the DNA’s stable helical form shields the bases that are susceptible to reduction, it is unable to generate an electrochemical reaction on its own [43]. A positive shift in the anodic and cathodic peak potential implies intercalation, while a negative shift suggests non-intercalation, i.e., electrostatic, hydrophobic, or groove binding. Another parameter, i.e., formal potential calculated as the average of anodic and cathodic peak potentials was also used in this regard to get an idea about the mode of interaction. For the intercalative mode of interaction of the compound with DNA, the formal potential shifts positively, while for the non-intercalative mode of interaction, the formal potential shifts negatively [44]. Here, very clear positive shifts given in Appendix A are observed in the electrochemical potential of the redox couple, independent oxidation, and reduction potentials as well as in the formal potential. Therefore, it may be deduced from the foregoing that the complexes interact with SS-DNA by means of intercalation. Voltammograms show that along with these alterations comes a reduction in current intensity. The development of heavy complex-DNA adducts, which diffuse slowly towards the electrode surface, is blamed for this shift because it lowers the concentration of the species responsible for the electrochemical reaction [44].

#### 2.5.3. DNA Interaction Study through Viscometry

The interaction of SS-DNA with ligand acids and their heteroleptic Cu(II) carboxylates was also evaluated through viscometry, which is a hydrodynamic method capable of providing valuable information about interaction mode. Each kind of compound–DNA interaction induces its own specific hydrodynamic response. This method is sensitive to change in chain length which is directly relevant to the interaction mechanism. An intercalating compound sits in between the base pairs of DNA and causes separation and unwinding of the DNA double helical structure. This finally results in an increase in the viscosity of the SS-DNA [45]. The viscosity SS-DNA shows a gradual increase with the increasing concentrations of ligand acids **L^1^**, **L^2^** and their heteroleptic Cu(II) carboxylates indicating the existence of intercalation [46]. The representative graph revealing this kind of changes is presented in Figure 7, strongly supporting the findings from the absorption and electrochemical methods, whereas those for ligand **L^2^** and complexes **4**–**6** are given in Appendix A.

### 2.6. Antioxidant Activities

#### 2.6.1. DPPH Scavenging Ability

The antioxidant potential of the free acids and their Cu(II) carboxylates was explored via the DPPH method using ascorbic acid as a reference. The DPPH, with a single unpaired electron, is capable of accepting a hydrogen or electron and gives a strong absorption at 517 nm. The absorption generally decreases when it accepts a hydrogen or electron from an antioxidant, forming a stable molecule. The results given in Table 5 indicate that the compounds under study show a moderate level of antioxidant potential. This activity may be attributed to the various structural and electronic factors like coordination geometry, redox properties, chelate ring size, degree of unsaturation, etc. [47]. The transfer of proton or electron from an antioxidant to DPPH results in the formation of species that undergo other reactions like coupling, fragmentation, and addition, which affect the rate and stoichiometry of the reaction. This leads to change in color from violet to yellow and hence a decrease in absorption at 517 nm which can be monitored UV-Visible spectrophotometer and the data indicate a direct activity concentration relationship [48]. The highest activity was observed for complex **2**.

#### 2.6.2. Hydrogen Peroxide Scavenging Ability

In addition to DPPH the ligand acids and their complexes were also evaluated for their peroxide scavenging ability test. Hydrogen peroxide (H_2_O_2_) is a biologically important, non-radical reactive oxygen species (ROS) that can influence several cellular processes. It is a weak oxidizing agent and can inactivate a few enzymes directly, usually via oxidation of essential thiol (-SH) groups. It can cross cell membranes rapidly, and inside the cell, H_2_O_2_ probably reacts with Fe^2+^ (and possibly Cu^2+^) ions to form the hydroxyl radical, which may be the origin of many of its toxic effects. It is, therefore, biologically advantageous for cells to control the amount of hydrogen peroxide that is allowed to accumulate [49]. A response was noted in terms of percentage scavenging ability and IC_50_ values and is presented in the Table 5. The data presented here indicate a direct relationship between the concentration of the antioxidant (compounds in the present study) and the percentage scavenging ability. The response in percentage ranges from 51.90 ± 1.16 to 87.63 ± 0.64 a little lower to compared to that of the reference used in the study. Again, the maximum activity was recorded for complex **2** followed by complex **3**.

### 2.7. Enzyme Inhibition Study

#### Acetylcholinesterase Inhibition

The synthesized compounds were evaluated for acetylcholinesterase (ACh) and butyrylcholinesterase (BCh) enzyme inhibitory activities. They were tested at five different concentrations (1000, 500, 250, 125, and 62.5 μg/mL) displayed varying degree of inhibition of enzyme acetylcholinesterase. The data are given in Table 6 for presenting percentage inhibition and IC_50_ values. The ligand acids **L^1^** and **L^2^** were found to exhibit a lower degree of inhibition compared to the complexes and the standard drug glutamine used here as a reference. The complex **2** showed a highest percentage inhibition at the concentration of 1000 μg/mL with IC_50_ 2 μg/mL. A drop in the activity was observed as the concentration was reduced from 1000 μg/mL to 500 μg/mL, 250 μg/mL, 125 μg/mL, and then to 62.5 μg/mL. The other complexes also showed their effect in a similar fashion with varying degree of inhibition.

Similar behavior has also been observed for the compounds under study when they were subjected to inhibition study of butyrylcholinesterase enzyme (Table 7). Enzyme inhibition and the concentrations of inhibitors were found to have a direct relationship. However, they all possess a lower degree of activity compared to the standard drug used as reference. The order of activity was found to be **4** > **5** = **2** > **1** > **3** > **6**, which may be attributed to various structural and electronic factors. Similar reports are available for the previously reported Cu(II) complexes as per the cited data [50].

## 3. Experimental

### 3.1. Materials and Instruments Used

The following chemicals were used: 2-chlorophenyl acetic acid (**L^1^**), 3-chlorophenyl acetic acid (**L^2^**), sodium bicarbonate, copper sulphate pentahydrate, 3-cyanopyridine, 4-cyanopyridine, and 4-hydroxypyridine were acquired from Fluka, Switzerland. Sodium salt of Salmon fish sperm DNA (SS-DNA) was purchased from Arcos, UK and was used as received. Analytical grade solvents were used as such. The following instruments were used: Electrothermal Gallenkamp (UK) serial number C040281 for melting point determination, Thermo Nicolet-6700 spectrophotometer for recording FTIR spectrum (4000–400 cm^−1^), Shimadzu 1700 UV-Visible spectrophotometer for absorption measurement, Corrtest CS 300 electrochemical workstation for electrochemical behavior study.

### 3.2. Single Crystal XRD Analysis

Diffraction data were collected by the ω-scan technique, with two Rigaku four-circle diffractometers: SuperNova (Atlas CCD detector) for complex **2**, at 130(1) K with a mirror-monochromatized CuKα radiation source (λ = 1.54178 Å), and XCalibur (Eos CCD detector) for **5**, at 100(1) K with graphite-monochromatized MoK_α_ radiation source (λ = 0.71073 Å). The data were corrected for Lorentz-polarization as well as for absorption effects [51]. The structures were solved with SHELXT [52] and refined by a full-matrix least-squares procedure on *F*^2^ employing SHELXL-2013 [53].

### 3.3. Synthesis of 2-Chlorophenyl Acetic Acid-Based Complexes ***1**–**3***

10 mL of an aqueous solution of NaHCO_3_ (4.2 mg, 5 mmol) was added dropwise to the 10 mL solution of 2-chlorophenyl acetic acid (8.5 mg, 5 mmol) in doubly distilled water under constant stirring for 3–4 h at 25 °C to convert the ligand into its sodium salt. After that, the temperature was raised to 60 °C and 10 mL of CuSO_4_·5H_2_O_(aq)_ (6.2 g, 2.5 mmol) and 10 mL of 2.5 mmol methanolic solution of nitrogen donor heterocycle 3-cyanopyridine were added dropwise simultaneously and the reaction was further stirred for 3–4 h (complex **1**). The same synthetic procedure was used for 4-cyanopyridine (complex **2**) and 4-hydroxy pyridine (complex **3**) (Figure 1). The precipitated products were washed with distilled water and then air dried. The equimolar solution (1:1) of DMSO and methanol was used for recrystallization of the product [22]. The physical and FTIR data are given in Table 7.

#### Synthesis of 3-Chlorophenyl Acetic Acid-Based Complexes **4**–**6**

Similar procedure as described above (Figure 1) was used for the synthesis of complexes **4**–**6**. The only difference is the use of 3-cholorophenyl acetic acid as a ligand in the first step of the synthesis.

### 3.4. Compound-DNA Interaction Study

The interaction ability of the ligand acids (**HL^1^**^−**2**^) and their synthesized Cu(II) carboxylates (**1**–**6**) was explored with SS-DNA. This study was carried out through UV-Visible absorption spectroscopy, viscometry, and cyclic voltammetry.

### 3.5. Compound-DNA Interaction Study through UV-Visible Absorption Spectroscopy

20 mg of SS-DNA was dissolved in 25 mL of double distilled water and the solution was left on stirring for 24 h at room temperature. Dilution of this stock solution was carried out and the final concentration was found to be 1.06 × 10^−4^ M using the molar absorptivity ε = 6600 M^−1^ cm^−1^, λ = 260 nm, and the path length of cell, *l* = 1 cm. The nucleotide to protein ratio calculated by using the absorbance at A_260_/A_280_ nm was found to be ∼1.7, indicating that the solution is certainly free from protein. The solutions of the ligand acids and the complexes under study were made in ethanol having a concentration of 1 mM. The experiment was carried out by adding DNA to a constant concentration of compound in increments. During the experiment, identical amounts of DNA were introduced to both the reference and sample cells in order to neutralize the effect of DNA absorption [29].

### 3.6. Voltammetry-Based Analysis for Compound-DNA Interactions

Cyclic voltammetry was used to confirm the compound-DNA interaction study, and a Corrtest CS 300 (Potentiostat/Galvanostat) electrochemical workstation with a glassy carbon working electrode (diameter = 0.03 cm^2^), a platinum wire working electrode, and a silver/silver chloride (Ag/AgCl) reference electrode were used in a continuous flow of argon. A glassy carbon electrode was polished on a nylon buffing pad with alumina and distilled water before each test to remove any absorbed contaminants. The cyclic voltammograms were taken when DNA was added in small increments while the quantities of the tested compounds were held constant [54].

### 3.7. Viscometry-Based Analysis for Compound-DNA Interactions

The purpose of the viscometric investigation was to document the shift in SS-DNA viscosity in response to the compounds. Ubbelohde viscometers were used to time the flow of DNA with and without ligand acids and their complexes and recorded using a digital stopwatch. The average reading was noted by repeating the experiment three times. Value of η_o_ was calculated by subtracting the flow time of pure solvent ethanol (t_o_) from that of the SS-DNA solution (t). The η was determined by comparing the flow rates of pure DNA solution (t) and solutions containing various concentrations of compounds (t′) in order to determine the effect of the compounds on the DNA solution flow rate. The graphs were drawn using the data $\sqrt[{3}]{\eta /\eta_{o}}$ vs. (compound)/(DNA) [29].

### 3.8. Antioxidant Activity

The synthesized compounds were subjected to DPPH (2,2-diphenyl-1-picrylhydrazyl) and H_2_O_2_ free radical scavenging ability test through procedures described in the following sections in order to get an idea about the antioxidant ability.

#### 3.8.1. DPPH Scavenging Assay

The compounds under study were subjected to DPPH scavenging ability test as per cited literature [55]. A 0.004% solution of reagent DPPH was added to the different concentrations of tested compounds (i.e., 125, 250, 500 and 1000 μg/mL) and the reaction mixture was incubated subsequently for about thirty minutes in dark. Ascorbic acid was used as positive control. The UV-3000 O.R.I. Germany was used to record the change in absorption of the reaction mixture at 517 nm and the percentage scavenging ability of the compounds under study was determined using formula:$$\%\;\mathrm{scavenging}\;\mathrm{activity}=\frac{\mathrm{Absorbance}\;\mathrm{of}\;\mathrm{control}-\mathrm{Absorbance}\;\mathrm{of}\;\mathrm{compounds}\;}{\mathrm{Absorbance}\;\mathrm{of}\;\mathrm{control}}\times 100$$

The experiments were repeated thrice. The GraphPad Prism^®^ (version 4.0, Sandiego, CA, USA) was used to calculate the IC_50_ values.

#### 3.8.2. Hydrogen Peroxide Scavenging Assay

The ligand acids and their Cu(II) carboxylates were further subjected to H_2_O_2_ scavenging ability test potential by following the procedure as per cited literature [56]. A 2 mM solution of H_2_O_2_ was made in 50 mM phosphate buffer having a pH 7.4. In the next step, 0.1 mL of the screened compounds was added to the 0.3 mL (50 mM) of phosphate buffer, then 0.6 mL of H_2_O_2_ was added and the solution was vortexed. After following the incubation period of 10 min, the absorption was 230 nm in comparison to the blank. Later on, these data were used to calculate the H_2_O_2_ free radical scavenging ability by applying the following equation:$$\mathrm{Hydrogen}\;\mathrm{peroxide}\;\mathrm{Scavenging}\;\mathrm{ability}=\frac{1-\mathrm{absorbance}\;\mathrm{of}\;\mathrm{sample}}{\mathrm{Absorbance}\;\mathrm{of}\;\mathrm{control}}\times 100$$

### 3.9. Enzyme Inhibition Study

The enzymes acetylcholinesterase (AChE) and butyrylcholinesterase (BChE) play an important role in the transfer of signals and physiological function. They assist acetylcholine to hydrolyze and produce choline and acetyl group in synaptic region. So, they are considered as targets in the management of Alzheimer’s disease. Herein, the compounds under study were evaluated for their potential to inhibit acetylthiocholine iodide (AChI) and butyrylthiocholine iodide (BChI) enzymes.

The well-known Ellman’s assay [57] was implemented to evaluate the inhibitory potential using acetylthiocholine iodide (AChI) and butyrylthiocholine iodide (BChI) as substrates, respectively. The basic principle of the assay is the hydrolysis of acetylthiocholine iodide and butyrylthiocholine iodide by their corresponding enzymes, resulting in the formation of 5-thio-2-nitrobenzoate anion. This anion is capable of forming a yellow color complex with 5,5-dithio-bis-(2-nitrobenzoic acid (DTNB), which shows absorption at 412 nm.

To carry out the assay, 0.1 M buffer solution with pH 8 was prepared as per the cited literature [58] where the pH was adjusted using KOH (potassium hydroxide). Using the freshly prepared buffer and following the dilution, final concentrations of 0.03 U/mL for AChE (518 U/mg solid) and 0.01 U/mL for BChE (7–16 U/mg) were obtained. Similar dilutions in methanol were also prepared for the galantamine which was selected as positive control. After that, the final solution of each of the AChE and BChE was prepared in distilled water in the presence of 2.27 × 10^4^ M DTNB and were stored at 8 °C. The experiment was performed by taking 205 µL of inhibitor (tested compound) along with 5 µL of prepared solutions of enzymes, followed by the addition of 5 μL DTNB reagent. They were then incubated for about 15 min in a water bath at a temperature of 30 °C. Later on, 5 µL substrate solution was added to them which were subjected to absorption check at 412 nm. A 10 μg/mL galantamine was used as a positive control, whereas the other components in the solution other than the inhibitor acted as a negative control. The temperature of the spectrophotometer was adjusted at 30 °C and then, following the reaction time of 4 min, the absorbance values were noted after regular intervals. The experiment was repeated, and change in absorption with time was used to calculate the percentage activity of enzyme and enzyme inhibitor [59]. The *p* values, or calculated probability levels, are categorized as: 5% (*p* < 0.05), 1% (*p* < 0.01) and 0.1% (*p* < 0.001). *p* < 0.05 means statistically significant and *p* < 0.001 means highly statistically significant.

## 4. Conclusions

Six new heteroleptic Cu(II) carboxylates were synthesized by using 2-chloro phenyl acetic acid and 3-chloro phenyl acetic acid as primary ligands and the substituted pyridine derivatives as auxiliary ligands. The complexes were characterized through FTIR spectroscopy which reveals the bridging bidentate mode of coordination for the ligands. Additionally, UV-visible absorption spectroscopy indicates the involvement of π-π* transitions. The cyclic voltammograms indicate that the complexes are redox active compared to the ligands, which were silent electrochemically. The complexes **2** and **5** were also characterized through single crystal XRD and both the complexes are dinuclear and adopt a distorted square pyramidal geometry where each carboxylate moiety presents a bridging bidentate coordination with nitrogen donor auxiliary ligand sitting at the terminal position. The ligand acids and complexes were evaluated for their interaction study with SS-DNA through UV-visible absorption spectroscopy, cyclic voltammetry, and viscometry. The findings from each study support each other with the conclusion that all of the compounds interact with DNA through intercalation with a high binding affinity through a spontaneous process as determined by the negative ΔG value. The complexes were also found to exhibit a moderate level of antioxidant potential when tested for the DPPH and H_2_O_2_ free radical scavenging ability. They were found to be potent inhibitors of acetylcholinesterase and butyrylcholinesterase.

## Data Availability

Data will be available on request to corresponding authors.

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
