# Peer review of "Synthesis, Structural Elucidation and Pharmacological Applications of Cu(II) Heteroleptic Carboxylates"

_pharmaceuticals, 2023, doi:10.3390/ph16050693_

Round 1
Reviewer 1 Report
This manuscript describes the preparation of six heteroleptic Cu(II) carboxylate complexes possessing 2-chlorophenyl acetic acid or 3-chlorophenyl acetic acid and three different substituted pyridine ligands. The complexes and their interaction with ss-DNA were characterized by several techniques including FT-IR, cyclic voltammetry, viscometry-based analysis, DPPH and H2O2 scavenging assays, and enzyme inhibition study, as well as two complexes 2 and 5 were characterized by single crystal XRD. The authors showed that all six complexes interact with ss-DNA through intercalation with a high binding affinity and exhibited a moderate level of antioxidant, as well as can be potent inhibitors of acetylcholinesterase and butyrylcholinesterase. The topic is suitable for publication in Pharmaceuticals. Below is a list of revisions that should be addressed to improve the quality of the manuscript before it is accepted.
1. Formatting: The references should follow the Pharmaceuticals reference style.
2. Formatting: There are various section 1.1 in this manuscript, section renumbering is needed.
3. Line 59: What does “this” in the “This is a major problem…” refers to? And suitable reference(s) should be cited to support the statement.
4. Line 64-65: Suitable references should be cited to support the statement “however, the already-in-use drugs are suffering from side 64 effects and selective activities.”
5. Line 79-81: Suitable references should be cited to support the statement “With the ability to adopt various easily accessible oxidation states, it is part of many enzymes involved in important biochemical processes in mammalian cells.”.
6. Line 103: “NASIDs” should be “NSAIDs” and the full name should be mentioned before the use of this abbreviation.
7. Lines 128-136: Why did the author choose only complexes 2 and 5 for single crystal XRD analysis?
8. Lines 138-148: Detailed information about the amount of solvent or solution concentrations is needed to provide for the solution of NaHCO3, 2-chlorophenyl acetic acid, 3-chlorophenyl acetic acid, and methanolic solutions of all nitrogen donor heterocycles.
9. Table 2, Column 2: The formula of compounds # 5 and 6 are incorrect according to information provided by Scheme 1.
10. Table 2: The author should separate current Table 2 into two different tables with one for FTIR data (the current bottom part of Table 2)
11. Lines 156-159: More information about ss-DNA (sources, types,…) should be provided.
12. Line 191: The full name of “DPPH” should be mentioned before the use of this abbreviation.
13. Line 227: The full name of “DTNB” should be mentioned before the use of this abbreviation.
14. Lines 246-247: Specific yields should be provided for all six heteroleptic Cu(II) carboxylate complexes synthesized.
15. Line 352: reformatting (section numbering) is needed here.
Minor editing of English language required
Author Response
We highly appreciate the effort of the honorable reviewer who reviewed our paper very critically and refined our paper by pointed out the below mentioned recommendations which are endorsed in the text (yellow highlighted).
- Formatting:The references should follow the Pharmaceuticals reference style.
Author Response: We have now downloaded the MDPI endnote reference style and all the references are now according to the Pharmaceuticals reference style.
- Formatting:There are various section 1.1 in this manuscript, section renumbering is needed.
Author Response: We apologize for the automatic numbering error which is now corrected.
- Line 59:What does “this” in the “This is a major problem…” refers to? And suitable reference(s) should be cited to support the statement.
Author Response: Since the paragraph is about the enzymatic activity of Cu(II) carboxylates against Alzheimer Disease particularly. So “this” refers to Alzheimer Disease. Reference #5-6 are cited.
- Line 64-65:Suitable references should be cited to support the statement “however, the already-in-use drugs are suffering from side effects and selective activities.”
Author Response: Reference #7 is cited.
- Line 79-81:Suitable references should be cited to support the statement “With the ability to adopt various easily accessible oxidation states, it is part of many enzymes involved in important biochemical processes in mammalian cells.”
Author Response: Reference #11 is now cited.
- Line 103:“NASIDs” should be “NSAIDs” and the full name should be mentioned before the use of this abbreviation.
Author Response: Nonsteroidal anti-inflammatory agents (NSAIDs) is now corrected as suggested.
- Lines 128-136:Why did the author choose only complexes 2 and 5 for single crystal XRD analysis?
Author Response: We have tried all the prepared complexes for crystallization in various solvents but crystals of only 2 and 5 were formed.
- Lines 138-148:Detailed information about the amount of solvent or solution concentrations is needed to provide for the solution of NaHCO3, 2-chlorophenyl acetic acid, 3-chlorophenyl acetic acid, and methanolic solutions of all nitrogen donor heterocycles.
Author Response: Detail is now mentioned in the text.
- Table 2, Column 2:The formula of compounds # 5 and 6 are incorrect according to information provided by Scheme 1.
Author Response: Thanks for identify the critical mistake. It is now corrected
- Table 2:The author should separate current Table 2 into two different tables with one for FTIR data (the current bottom part of Table 2).
Author Response: Done as suggested
- Lines 156-159:More information about ss-DNA (sources, types,…) should be provided.
Author Response: Sodium salt of Salmon fish sperm DNA (SS-DNA) was purchased from Arcos, UK and was used as received. Detail is now also mentioned in the manuscript.
- Line 191:The full name of “DPPH” should be mentioned before the use of this abbreviation.
Author Response: Full name of DPPH (2,2-diphenyl-1-picrylhydrazyl) is now mentioned in the text as suggested.
- Line 227:The full name of “DTNB” should be mentioned before the use of this abbreviation.
Author Response: Full name of DTNB (5,5-dithio-bis-(2-nitrobenzoic acid)) is now mentioned in the text as suggested.
- Lines 246-247:Specific yields should be provided for all six heteroleptic Cu(II) carboxylate complexes synthesized.
Author Response: Yields are mentioned in Table 2.
- Line 352:reformatting (section numbering) is needed here.
Author Response: Done according to the format of the journal as suggested.
Reviewer 2 Report
The authors present their work on the synthesis of 6 novel Cu(II) heteroleptic carboxylates, their characterization, and testing for binding to DNA, inhibition activity against two cholinesterase enzymes, and scavenging ability towards DPPH and H2O2. The work is clearly presented (although some suggested improvements are provided below) and results seem valid. I would suggest some improvements to the paper:
1. Scheme 1 is difficult to decipher. Some compound structures (for the two ligands and for the pyridines) should be provided to make it easier for the reader.
2. Table 2 has incorrect formulas for compounds 5 and 6 (wrong pyridines).
3. Line 202 "y" is missing from activity and line 214: only one "l" in control.
4. Section 1.1: crystal data is provided for complex 1, but the x-ray structures for 2 and 5 were the only ones provided.
5. Figure 3: solvent should be included. Why not include a 2nd plot for the other complexes as well?
6. Line 322: I believe current intensity changes with scan rate (increase in current with increase in scan rate). I think this has nothing to do with whether the electrochemical process is irreversible.
7. The CV discussion is difficult to follow in the absence of seeing a CV for the pyridines alone as well as where the Cu(I)/Cu(II)/Cu(III) couples occur when not complexed. I would suggest including this data.
8. Figure 4: scan rates need to be included with all CV scans.
9. Lines 380-382. This is rather speculative as the authors have no idea whether these complexes will be effective against a disease given only the fact that they found that they intercalate.
10. Line 387. By "planner", do the authors mean "planer"?
11. Line 393: need to include "incremental addition of DNA"
12. Line 407-409: a reference is needed for this suggestion.
13. Figure 6 title: "Cyclic" and "indicating" need to be corrected
14. Line 445: I believe the authors mean that DPPH will accept a hydrogen atom (not a proton, H+)
15. Table 5: the authors need to describe what the values in the table refer to. They are described in the experimental section, but some description needs to be included in the text for the reader to understand the values. Something along the lines of "a higher value means ...." would be helpful as it would put the values into a frame of reference for the reader. A similar statement putting the values in Table 6 into a frame of reference would be welcome.
16. Line 485: I believe it is supposed to read "was reduced from 1,000..." (not 100).
With attention to these changes, I believe this paper would be publishable.
The english is good with attention to the suggestions above.
Author Response
Dear Reviewer thanks a lot of your precious time spared for our paper. We have tried to the best of our level to incorporate all the suggestions and corrections. The corrections are yellow highlighted in the text.
- Scheme 1 is difficult to decipher. Some compound structures (for the two ligands and for the pyridines) should be provided to make it easier for the reader.
Author Response: Structure of complexes 2 and 5 are now included in the Scheme 1 as the remaining are differ only by the position of substituent at pyridine ring.
- Table 2 has incorrect formulas for compounds 5 and 6 (wrong pyridines).
Author Response: Thanks for identify the critical mistake. It is now corrected.
- Line 202 "y" is missing from activity and line 214: only one "l" in control.
Author Response: Corrections in both equations are now done.
- Section 1.1: crystal data is provided for complex 1, but the x-ray structures for 2 and 5 were the only ones provided.
Author Response: The mistake is rectified. The crystal data is for complexes 2 and 5 only.
- Figure 3: solvent should be included. Why not include a 2nd plot for the other complexes as well?
Author Response: Ethanol solvent was used which is now mentioned in the figure caption. The plot for others is provided as well in supplementary data, Figure S1.
- Line 322: I believe current intensity changes with scan rate (increase in current with increase in scan rate). I think this has nothing to do with whether the electrochemical process is irreversible.
Author Response: Yes the comment is justified, so the statement is removed to avoid further confusion. The linear change in potential and current with increasing scan rates is indicative of the diffusion controlled and an electrochemically feasible process.
- The CV discussion is difficult to follow in the absence of seeing a CV for the pyridines alone as well as where the Cu(I)/Cu(II)/Cu(III) couples occur when not complexed. I would suggest including this data.
Author Response: Our focus was to check the change in electrochemical behavior of synthesized complexes without going into the deep structural elucidation through CV. So we did not record the data individually for reacting moieties. But the in details in manuscript along with cited literature may provide help.
- Figure 4: scan rates need to be included with all CV scans.
Author Response: Done as suggested.
- Lines 380-382. This is rather speculative as the authors have no idea whether these complexes will be effective against a disease given only the fact that they found that they intercalate.
Author Response: The speculation is made on the fact that lot of natural and synthesized complexes are part of effective medicine for various diseases just because of the fact that they are good DNA intercalator.
- Line 387. By "planner", do the authors mean "planer"?
Author Response: The spelling mistake is rectified and yes it is planer.
- Line 393: need to include "incremental addition of DNA"
Author Response: The concentrations of DNA (0.49, 0.99, 1.48, 1.96 µM) addition is now mentioned in the text as suggested.
- Line 407-409: a reference is needed for this suggestion.
Author Response: Reference# 54 is now added.
- Figure 6 title: "Cyclic" and "indicating" need to be corrected.
Author Response: Done.
- Line 445: I believe the authors mean that DPPH will accept a hydrogen atom (not a proton, H+)
Author Response: Yes hydrogen atom not a proton. Correction is done.
- Table 5: the authors need to describe what the values in the table refer to. They are described in the experimental section, but some description needs to be included in the text for the reader to understand the values. Something along the lines of "a higher value means ...." would be helpful as it would put the values into a frame of reference for the reader. A similar statement putting the values in Table 6 into a frame of reference would be welcome.
Author Response: The P values, or calculated probability levels are categorized as: 5% (P < 0.05), 1% (P < 0.01) and 0.1% (P < 0.001). P < 0.05 means statistically significant and P < 0.001 means statistically highly significant. It is now mentioned in the text also.
- Line 485: I believe it is supposed to read "was reduced from 1,000..." (not 100).
Author Response: Yes it is 1000. We have corrected the typing mistake.
Round 2
Reviewer 2 Report
I appreciate the author's attention to my previous recommendations. I have to correct myself in that on line 264, "planer" should be "planar". With that correction, I support publication of the paper.